# CD37 in B Cell Derived Tumors—More than Just a Docking Point for Monoclonal Antibodies

**DOI:** 10.3390/ijms21249531

**Published:** 2020-12-15

**Authors:** Malgorzata Bobrowicz, Matylda Kubacz, Aleksander Slusarczyk, Magdalena Winiarska

**Affiliations:** Department of Immunology, Medical University of Warsaw, 02-097 Warsaw, Poland; malgorzata.bobrowicz@wum.edu.pl (M.B.); matyldakubacz@gmail.com (M.K.); slusarczyk.aleksander@gmail.com (A.S.)

**Keywords:** CD37, tetraspanin, non-Hodgkin lymphoma, chronic lymphocytic leukemia, immunotherapy, monoclonal antibodies, CD20

## Abstract

CD37 is a tetraspanin expressed prominently on the surface of B cells. It is an attractive molecular target exploited in the immunotherapy of B cell-derived lymphomas and leukemia. Currently, several monoclonal antibodies targeting CD37 as well as chimeric antigen receptor-based immunotherapies are being developed and investigated in clinical trials. Given the unique role of CD37 in the biology of B cells, it seems that CD37 constitutes more than a docking point for monoclonal antibodies, and targeting this molecule may provide additional benefit to relapsed or refractory patients. In this review, we aimed to provide an extensive overview of the function of CD37 in B cell malignancies, providing a comprehensive view of recent therapeutic advances targeting CD37 and delineating future perspectives.

## 1. Introduction

CD37 is a tetraspanin expressed especially on the surface of B cells. Tetraspanins, the so-called transmembrane-four superfamily proteins (TM4SF), are a large family of evolutionarily conserved proteins present on the cell surface and intracellular vesicles [1]. This family of membrane-organizing proteins modulates signal transduction with a myriad of molecules and controls several fundamental processes, i.e., cell motility, proliferation, and cytoskeletal reorganization; consequently emerging as an important player regulating the mechanisms of the immune response [2,3]. CD37 molecule is an excellent example of tetraspanins, described for the first time by Link et al. [4], which has recently been perceived as a promising target for research in immunotherapy [5,6,7]. This antigen is a heavily glycosylated protein prominently abundant on the surface of both healthy and malignant B cells. Owing to additional carbohydrate chains, the molecular mass of CD37 fluctuates from approximately 40 to 64 kDa [4,8]. A mounting body of research demonstrates that CD37 is crucial in regulating B cell survival, shaping immune response, and immune evasion [2]. It seems that CD37 constitutes more than a docking point for monoclonal antibodies, and its targeting with immunotherapy may provide additional benefit for refractory patients with B cell malignancies. Therefore, this review aims at summarising the role of CD37 in B cell malignancies, providing a comprehensive view of recent therapeutic advances targeting CD37 and delineating future perspectives.

## 2. Characteristics of Tetraspanins as Immune Regulators

It is worth noticing that although all tetraspanins cross the cell membrane four times, not all proteins that cross it four times are tetraspanins [9]. The tetraspanin family that consists of 33 members in humans has a well-defined structure comprising a short amino- and carboxy-terminal tail, three transmembrane regions (TM1, TM2, and TM3) divided by small intracellular and small and large extracellular loops [9]. A conserved signature CCG motif (a Cys–Cys–Gly sequence) on the large extracellular loop is a hallmark of tetraspanin proteins (see [9,10] for details).

Several tetraspanins are broadly distributed (CD9, CD63, CD81, CD82, and CD151), while the expression of others is highly restricted to hematopoietic cells (CD37, CD53, Tssc6, and Tspan33) [2]. In general, these proteins form complexes with other members of the tetraspanin family and with a plethora of cytosolic, transmembrane, and surface proteins such as integrins, growth factor receptors, immune receptors (CD4, CD8, CD19, and lymphocyte antigen receptors), cytosolic signal transduction molecules (phosphatidylinositol 4-kinase, protein kinase C), G-protein-coupled receptors, immunoglobulin superfamily members, and cytoskeletal components [11,12].

Briefly, tetraspanins are thought to be modulators of signal transduction, providing organization to membrane domains through lateral interactions with their partners [2,13]. Due to their indirect involvement in signal transduction, they are generally described as “molecular facilitators” of signal transduction [14]. Depending on the cell type, tetraspanins interact with specific partner molecules. Moreover, by coupling to cholesterol and gangliosides, they form higher-order tetraspanin complexes, the so-called tetraspanin-enriched membrane microdomains (TEMs), that function as signaling platforms [9], places of dynamic interaction that rapidly exchange proteins within the cell membrane [15]. These protein-protein interactions are executed mainly by a reversible lipid modification—palmitoylation of intracellular, juxtamembrane cysteines [16,17].

The importance of tetraspanins in controlling immunity has been demonstrated in numerous tetraspanin deficient murine models that reported their involvement in humoral and/or cellular immune responses (see [2] for details). Tetraspanins participate in a wide array of cellular processes, including cell adhesion, motility, differentiation, proliferation, and tumor invasion, as elegantly described in [1,2,9,12] (Figure 1). Here we aim at briefly summarising their involvement in controlling immunity.

As already mentioned, immune cell markers and receptors (CD4, CD8, CD19, T cell receptors (TCRs), and B cell receptors (BCRs)), integrins such as integrin β1, β2, or β7 and a range of signaling molecules, e.g., PI4-kinase, PKC, and growth factors with their receptors [18,19] have been shown to interact with specific proteins of TM4SF. These interactions lead to modulation of signal pathways underlying plentiful cellular processes such as antigen presentation, cell proliferation, antibody production, and many more [11,12]. A significant illustration of tetraspanin-integrin connections is a regulation of β2 integrin-mediated adhesion and migration in neutrophils mediated by CD37. It was observed that CXCL1-induced neutrophil adhesion and transmigration were reduced in CD37-deficient mice, which was followed by a reduced capacity to undergo β2 integrin-dependent adhesion [20].

Some of the TM4SF proteins also associate with MHC II on dendritic cells, playing a significant role in MHC clustering. [21]. Due to the ability of CD37 to interact with CD9, CD81, CD82 and MHC II [22,23], and it was thought that CD37 deficiency would dysregulate tetraspanin microdomains and consequently induce inhibition of MHC clustering with a poor presentation of antigens. However, the opposite effect was obtained as significantly higher stimulation of antigen-specific T cells was observed [24].

Regulation of peptide/MHC presentation could be explained by the fact that tetraspanins can exert both opposing and overlapping functions. Some of them, such as CD9 and CD82, can promote MHC clustering, yet others, such as CD37, sequestrates MHC and prevents the formation of complexes with other tetraspanins [25]. Lastly, tetraspanins play a pivotal role in endocytosis and trafficking of MHC molecules, as some of them can be internalized and further translocated to the endosomal–lysosomal pathway [26]. Due to the internalization process, many TM4SF proteins are distributed within intracellular organelles. CD37, for example, can be found on multilamellar endosomes in B cells [27].

Numerous tetraspanins (e.g., CD9, CD37, CD53, CD81, and CD82) are also involved in response to CD3-TCR engagement and fulfill costimulatory functions for T cells [22]. Data from murine models of targeted tetraspanin deficiency have demonstrated that CD37^−/−^, CD151^−/−^, Tssc6^−/−^, and CD82^−/−^ T cells are hyperproliferative upon TCR stimulation [28,29,30]. Importantly, the proliferation of CD37^−/−^ T cells is preceded by enhanced early IL-2 production [29]. It has been demonstrated that deficiency in CD37 leads to enhanced tyrosine kinase activity, known as CD4/CD8-associated Lck, essential for the TCR signal transduction pathway. Tetraspanins, including CD37, also play a crucial role in the trafficking of dendritic cells (DCs) to lymph nodes. DCs isolated from CD37-deficient mice showed decreased motility [25] and migration to lymph nodes [26]. Therefore CD37-deficiency in mice leads to impaired tumor rejection. Interestingly, another tetraspanin CD82 has an opposing function, which further underlines the importance of the tetraspanin web in controlling immune response mechanisms [25].

Tetraspanins are also engaged in the regulation of signaling pathways essential for macrophage activation. After recognition of certain components present on fungal cells, dectin-1 interacts with CD37 leading to the stabilization of this C-type lectin within the plasma membrane and inhibition of dectin-1-mediated IL-6 production by macrophages [31].

An important connection between tetraspanin functioning and pathology of infectious diseases has also been described, leading to a conclusion that within host cells’ membrane TM4SF family, there are proteins capable of interacting with microbial receptors [32,33]. In such a context, tetraspanins seem to fulfill the purpose of binding platforms for different types of pathogens, such as bacteria, viruses, or parasites [32].

In addition, the essential role of CD37 in B cell function and humoral immunity was reported [3]. CD37-deficient mice were characterized by insufficient T cell-dependent IgG responses as well as intensified IgA-mediated responses, which in turn resulted in spontaneous renal nephropathy mimicking IgA nephropathy [34]. Defective humoral immune responses provide evidence for the importance of CD37 in antibody production. CD37 is required for the binding of vascular cell adhesion molecule 1 (VCAM-1) to the α(4)β(1) integrin, which is crucial for activation of the Akt survival pathway. Lack of CD37 leads to increased apoptosis of plasma cells in the germinal centers of the spleen, which in turn results in reduced numbers of IgG-secreting plasma cells in lymphoid organs [35]. The impaired humoral immune response following CD37 deficiency is also due to the fact that CD37 seems to be an important player in providing costimulatory signals, but it is not indispensable for correct IgG production and B cell development [36].

Tetraspanins are also assumed to have a significant role in the control of cancers in humans [2]. It has been observed that expression levels of tetraspanins are associated with different types of tumors (reviewed in [37]) as well as prognosis of responses to treatment in patients [38]. As tetraspanins are thought to be major organizers of extracellular vesicles’ formation [39], they have been identified on the surface of exosomes present in metastatic tumors, expanding their role in providing mechanisms of immune evasion [39,40]. Several tetraspanins, including CD9, CD151, CD82 and Tetraspanin 8 have been described as members of extracellular vesicles (EVs). Although their exact role needs further elucidation, it seems plausible that they provide docking signals for endocytosis in the target cells. Thus targeting them may decrease the metastatic potential of several malignancies [41,42]. Moreover, it has been observed that a composition of the exosomal tetraspanin-complexes determines trafficking of exosomes to the target cells. Thus, it has been discussed that research aiming at identifying tetraspanins responsible for directing EVs to different cells may facilitate tailoring exosomes for drug delivery in the future [43].

## 3. CD37 in B Cell Lymphoma and Leukemia

Like other members of the tetraspanin family, CD37 has been extensively studied in the context of leukocyte biology and immunology [2,31,33,44,45]. On the surface of B cells, CD37 forms protein complexes with CD53, CD81, CD82, and class II glycoprotein [46]. As mentioned before, a large body of evidence on tetraspanin comes from tetraspanin-deficient mice. In general, mice with targeted inactivation of CD37 demonstrated no changes in the development of lymphoid organs [36]. As no abnormalities in B cell compartments have been observed, it is postulated that CD37 is dispensable for B cell development [36].

In contrast to normal cells’ development, it seems that CD37 is an important regulator of tumorigenesis in B cells. In fact, CD37 has been recently described as a negative regulator of tumorigenesis in B cell NHL. The crucial role of CD37 in B cell lymphoma formation comes from a murine model, where CD37-deficient mice spontaneously developed germinal center–derived B cell lymphoma in lymph nodes and spleens with a higher incidence than Bcl2 transgenic mice (50% vs. 20%) [45]. Of note, CD37^−/−^mice do not express increased amounts of Bcl2, meaning that lymphoma development is Bcl2 independent in this setting. In this study, CD37 has been discovered to interact with suppressor of cytokine signaling 3 (SOCS3), and its absence was hypothesized to drive lymphoma development through constitutive activation of the IL-6 signaling pathway (Figure 1). Importantly, mice lacking both CD37 and IL-6 were fully protected against lymphoma development, which confirms the involvement of the IL-6 pathway in driving tumorigenesis. The same study shows that loss of CD37 in neoplastic cells in patients with diffuse large B cell lymphoma (DLBCL) directly correlates with activation of the IL-6 signaling pathway and with worse progression-free and overall survival. Further observations revealed that CD37-negative DLBCL patients showed a significantly lower response and decreased survival after R-CHOP (rituximab+chemotherapy) treatment when compared to the CD37-positive population independently from CD20-status, International Prognostic Index (IPI), and cell of origin [47]. CD37 loss correlated with negative prognostic factors such as mutated TP53, high expression of NF-κB, Myc, phosphorylated STAT3 and survivin, p63 loss, and BCL6 translocation [47].

It has been suggested that CD37 may act as a “molecular facilitator” of rituximab action, especially for antibody-dependent cellular cytotoxicity, cross-linking, aggregation in lipid rafts, and tyrosine kinase transactivation, apoptosis induction, and long-term T cell responses [47]. These findings may explain a synergistic effect of targeting CD20 and CD37 by mAbs, as described further below. CD37 loss also correlated with decreased survival in the CD37-deficient Eµ-TCL1 mouse model of CLL. However, in terms of the time until the appearance of leukemia in peripheral blood, no significant difference was observed in CD37^−/−^ vs. wild type mice. Also, in line with the previous studies, non-TCL1 mice with CD37 deficiency did not exhibit impaired survival [48].

Based on the effect of SMIP-016, a CD37-specific small modular immunopharmaceutical that, upon cross-linking, induces apoptosis of CLL cells [49], it has been hypothesized that CD37 acts as a death receptor. However, given the presence of short cytoplasmic tails (8 to ~14 amino acids) that lack canonic signaling motifs, its direct role in the initiation of cell death seems unlikely. Nevertheless, a mass-spectrometry analysis revealed the presence of a weak immune tyrosine-based inhibitory motif (ITIM) at the N-terminal domain and a single immune tyrosine-based activation motif (ITAM) in the C-terminal domain [44]. These two tyrosine residues play opposing roles in cell survival (Figure 1). In experiments with SMIP-016, ligation of CD37 induced phosphorylation within the ITIM-like motif and association with a specific complex of proteins including LYN, SHP1, SYK, and PI3Kγ. A pro-apoptotic effect of ITIM ligation is thought to be mediated by SHP1 recruitment and FoxO3a-dependent *BIM* upregulation and subsequent mitochondrial depolarization and cell death. On the other hand, phosphorylation of the ITAM-like motif leads to recruitment and activation of a pro-survival pathway mediated by phosphorylation of PI3Kδ, AKT, and GSK3β. These findings have potential practical implications, as a combination treatment with SMIP-016 and the PI3Kδ isoform-specific inhibitor idelalisib has demonstrated efficacy in preclinical in vitro studies [44]. Similar observations have been made in the case of another chimeric anti-CD37 antibody-BI 836826, which has demonstrated increased efficacy against CLL cells in vitro when combined with PI3K inhibitor idelalisib [50].

Moreover, it has been suggested that CD37 may serve as a novel biomarker for anti-PD-1 blockage that is tested in clinical trials in DLBCL [47]. Together, these observations identify CD37 as a tumor suppressor that directly protects against B cell lymphomagenesis [44,45,47]. Recently, in DLBCL patients, it has been demonstrated that mutations leading to defective glycosylation and trafficking of CD37 that, in consequence, lead to the lack of CD37 on the cell membrane are present only in the population with immune-privileged site-associated tumors. Therefore, it has been suggested that CD37 loss provides a survival advantage in the otherwise stimulus-poor environment [51].

However, the observations from acute myeloid leukemia (AML) suggest that the role of CD37 in promoting or suppressing tumorigenesis may be tumor-dependent. CD37 mRNA expression was significantly upregulated in AML patients compared to healthy individuals. High CD37 expression in AML patients was associated with shorter overall survival and disease-free survival [52].

Interestingly, the very recent observations from non-small cell lung cancer (NSCLC) have suggested low expression of CD37 as a marker of metastasis. Gene expression profiling from fine-needle aspirates from NSCLC tumors has identified CD37 as one of three genes downregulated in the course of the disease progression [53]. Recent advances in tumor immunology demonstrate that tumor-infiltrating B cells influence tumor progression through the productions of antibodies, immunosuppressive cytokines such as IL-10, as well as by interacting with other immune cells [45]. Intriguingly, immunosuppressive IL-10 secreting B regulatory 1 (BR1) cells are known to support tumor growth have been shown to have downregulated CD37 (2 fold when compared to IL-10 non-secreting cells) [54]. The role of B cells in promoting tumor metastasis has been largely discussed, e.g., in breast cancer and NSCLC [54].

## 4. CD37 as a Molecular Target for Immunotherapy

CD37 was firstly described in 1986 as a molecular target for radioimmunotherapy with the aid of MB-1 antibody [4,55,56]. Despite promising results in both murine xenograft lymphoma model [57] and small clinical trials using a radiolabeled [^131^I] MB-1 anti-CD37 antibody [56,58], CD37 initially lost the battle with CD20 for a target used in immunotherapies. CD37 had been largely dimmed as a molecular target for monoclonal antibodies (mAbs) with the first approval of rituximab in B cell malignancies in 1997 [59].

Effectiveness of this anti-CD20 antibody against non Hodkin’s lymphoma world widely recognised as rituximab remains undisputed, especially as a first line in combination with chemotherapy. A significant improvement in response rate and survival was noticed in patients with B cell lymphoproliferative disease, which were positive in terms of CD20 [60,61,62,63,64]. It was noticed that rituximab’s mechanism of action relies on mediating complement-dependent cell lysis and antibody-dependent cellular cytotoxicity. Not only does it present such properties, but also can it sensitize chemoresistant cell lines leading to cell apoptosis. Unfortunately, there are still patients insusceptible to rituximab or the ones developing resistance to it as a consequence of CD20 downregulation [65]. The exact incidence of rituximab resistance in antibody-naive patients is a quite difficult issue to be analysed [66]. However, it was noticed, that after having been initially treated with rituximab, a re-treatment in case of relapse patients induces an overall response rate of only 40% [67]. Nonetheless, in 73% of patients a shrinkage of tumor was noticed equal to at least 20% [67]. Therefore, primarily due to the phenomenon of rituximab resistance, new treatment approaches are still required to be developed.

While anti-CD20 mAbs constitute a vital part of the booming market of immunotherapeutics, only a limited number of CD37-directed candidates have been evaluated in patients so far [68]. Nevertheless, recently, there has been broad interest in CD37′s revival as a therapeutic target [55,69]. It has been suggested that targeting CD37 with mAbs may be useful for patients resistant or refractory to anti-CD20 mAb therapy or relapsing after such treatment [68].

Targeting CD37 emerges as an additional opportunity also for the patients treated with kinase inhibitors [70], as CD37 has been demonstrated to couple with the PI3K-Akt survival pathway [44].

CD37, like its “rival” CD20, is absent on early progenitor cells or terminally differentiated plasma cells [4,8] and relatively highly expressed on malignant B cells, which makes it an ideal targets for the therapy of non-Hodgkin lymphoma (NHL) and chronic lymphocytic leukemia (CLL) [4,71]. Of note, the expression of CD37 is significantly higher on CLL cells than on normal peripheral B cells [72], and CLL cells express an even greater amount of CD37 mRNA than CD20 mRNA [73,74,75].

### 4.1. Anti-CD37 mAbs

There are numerous papers confirming the efficacy of monoclonal antibodies in B cell malignancies. However, research in this area is constantly expanding [76]. Currently, anti-CD37 mAbs assessed in clinical trials for the treatment of B cell malignancies include otlertuzumab (formerly known as TRU-016), naratuximab emtansine, AGS67E, BI 836,826, and betalutin. The main characteristics of these agents are presented in Table 1.

Otlertuzumab is a bio-engineered protein comprising anti-CD37 variable regions linked to an IgG constant domain, developed by humanizing its precursor agent, SMIP-016 [68,70,77]. SMIP-016 was demonstrated to induce apoptosis upon binding to CD37 [49]. Consequently, a pro-apoptotic protein BIM becomes upregulated, and a cascade of reactions lead to the cell’s programmed death [44]. Treatment of human B cell tumor lines with SMIP-016 increased apoptosis also in combination with chemotherapeutic drugs, rituximab, bendamustine, and rapamycin [78]. Preclinical studies conducted in vitro and in vivo provide evidence for otlertuzumab being both a mediator of apoptosis and a mediator of anibody dependent cellular cytoxicity (ADCC) against primary chronic lymphocytic leukemia and NHL cells [77]. Otlertuzumab has also shown efficacy in clinical trials in CLL [79].

Naratuximab emtansine (IMGN529) is an anti-CD37 monoclonal antibody conjugated to maytansinoid DM1. Upon binding of naratuximab emtansine with CD37, DM1 is internalized and released intracellularly, causing the disruption of microtubule assembly, cell cycle arrest, and apoptosis. Naratuximab emtansine demonstrated high anti-tumor activity when investigated in models of CLL and CD37-positive NHL in preclinical studies. Additionally, it showed significant ADCC activity towards lymphoma target cells and B cell lymphoma cell lines [80].

Another antibody-drug conjugate, AGS67E, was engineered by joining together a fully human monoclonal IgG2 antibody with the microtubule-disrupting agent monomethyl auristatin E (MMAE) via a protease-cleavable linker [74]. This agent induces potent cytotoxicity, apoptosis, and cell cycle alterations in NHL and CLL cell lines as well as primary cells in vitro. All the mentioned properties may render the developed antibody a promising and applicable immunotherapeutic in B/T cell malignancies therapies. What is more, AGS67E reveals anti-tumor activity in CLL and NHL xenografts, including rituximab-resistant cases. Furthermore, AGS67E may be a novel drug candidate in acute myeloid leukemia (AML) [74].

Another chimeric Fc-engineered anti-CD37 molecule with improved ADCC–BI 836826, successfully targets CLL cells [81], including a chemo-resistant side population, and has demonstrated increased efficacy when combined with PI3K inhibitor idelalisib [50]. A body of evidence suggests beneficial effects from a combination of anti-CD37 with anti-CD20 monoclonal antibodies. In preclinical studies, the efficacy of otlertuzumab has been potentiated by an anti-CD20 ofatumumab [82] in diffuse large B cell lymphoma (DLBCL) cell lines and CLL primary cells. Preclinical data from multiple B cell lymphoma cell lines have demonstrated a synergistic cytotoxic effect of an anti-CD37 naratuximab emtansine combined with anti-CD20 mAbs (rituximab, ofatumumab, and obinutuzumab) [83]. Also, data from xenograft DLBCL models show increased toxicity of the combination of naratuximab emtansine plus rituximab when compared to either agent alone or R-CHOP treatment [83]. It has been demonstrated that the observed synergy relies on the augmented internalization of anti-CD37 mAb following CD20 binding. However, the internalization of anti-CD20 mAbs remained unaffected following CD37 binding. The very recent observations have demonstrated that anti-CD20 and anti-CD37 mAbs form on the cell surface mixed hexameric (Hx) antibody complexes consisting of both antibodies, each bound to their own cognate target, so-called hetero-hexamers [7]. This study was conducted using a novel anti-CD37 mAb with potentiated complement-dependent cytotoxicity (CDC) efficacy. Introduction of a single point mutation-E430G in the IgG Fc domain enhanced intermolecular Fc-Fc interactions between cell-bound IgG molecules, thereby facilitating IgG hexamer formation [7]. This Hx-mAb has demonstrated effective CDC in a panel of B-NHL cell lines and primary CLL samples. Of note, it induced CDC activity even in Raji cells that were originally less susceptible to CDC because of CD59 expression. CD59 is a key regulator of the CDC that restricts the formation of membrane-attacking complex (MAC).

In an attempt to further improve the efficacy of CD37-targeting with Hx-mAbs, a DuoHexaBody-CD37 (GEN3009–a bispecific second-generation monoclonal antibody targeting two non-overlapping epitopes of CD37) with an E430G hexamerization-enhancing mutation has been generated [84]. DuoHexaBody-CD37 induces not only effective CDC against B cell lymphoma cell lines and primary CLL samples but also can induce efficient ADCC and antibody-dependent cellular phagocytosis (ADCP) in vitro by engaging FcγRs. The efficacy of DuoHexaBody-CD37 in repressing tumor growth has been demonstrated in xenograft models as well as in B-NHL patient-derived xenografts (PDX).

### 4.2. Chimeric Antigen Receptors (CARs) as Emerging Treatment Modality for Patients with Refractory Lymphomas

CD37 seems to be a rational target for chimeric antigen receptor (CAR)-modified cytotoxic cells. By now, two studies showed the efficacy of such a therapeutic concept comparable to the efficacy of CD19-directed CAR-T cells in controlling tumor growth and prolonging survival in xenograft models using human B cell lymphoma cell lines [85], as well as PDX models of NHL [75].

Throughout years of studies, remarkable clinical effects have been achieved with CAR-T cells in patients with relapsed or refractory B cell acute lymphoblastic leukemia (ALL) [86]. CD19 CAR-T cells treatment has also been proven greatly successful in B cell non-Hodgkin lymphoma patients. However, antigen-negative relapses have been observed following CD19 CAR-T cell therapy [87]. Therefore, there is an unmet need for improving the currently developed CAR-T cell therapies by searching for novel therapeutic targets. Consequently, an engineered construct targeting CD37 has been developed by two separate research teams, providing a glimpse into a possible future therapeutic approach to treatment [75,85].

The efficacy of CD19 CAR and CD37 CAR was compared in a study conducted by Köksal and Dillard et al. [85]. In this study, CAR-T cells targeted against CD37 were proven both specific and efficient against B cell lymphoma in in vitro studies and in two xenograft lymphoma models in mice. CD37 CAR- T cells were producing Th1-type cytokines (TNF-a, IFN-y, IL-2, and GM-CSF) and exerting cytotoxic properties. Importantly, CD37 CAR-T cells were more effective in terms of cytotoxic effect against U2932 (DLBCL) lymphoma cell line than CD19 CAR-T cells. In contrast to CD19 CAR, CD37 CAR-T cells were resistant to antigen masking, so tumor cells did not become resistant to it over time. Furthermore, available data presented no evidence of CD37 CAR- T cells toxicity against other peripheral blood immune cells.

Another attempt by *Scarfò* et al. resulted in the generation of two CARs against CD37 (CAR-37 L-H and CAR-37 H-L) differing from one another in orientations of variable heavy and light chains in scFv [75]. Proven to be similarly effective in in vitro studies using various B-NHL cell lines as anti-CD19, CAR-37 L-H, and CAR H-L were studied in a xenogenic model of mantle cell lymphoma (MCL), where they showed similar efficacy in the eradication of tumor cells. Moreover, the efficacy of CAR-37 H-L was tested in PDX models of MCL. Interestingly, CAR-CD37 T cells were able to eradicate tumor cells faster (after 12 days) compared to CAR-CD19 T cells (14 days, *p* < 0.05). Additionally, CAR-CD37 T cells were effective in inducing cell death in cutaneous T cell lymphoma (CTCL) cell lines characterized by a Th2 phenotype [75]. Given the successful application of bi-ligand immunoliposomes targeting CD19 and CD37 on B-CLL cells [88], tandem CD19/CD37 CAR-T cells were also generated [75]. CD19 CAR, CD37 CAR, and CD19/CD37 CAR were equally effective in tumor eradication in a xenograft MCL model. Nevertheless, bispecific CAR usage in lymphoma represents an attractive strategy to overcome the downregulation of the target molecules on lymphoma cells. Figure 2 provides an overview of the mechanisms of action of selected anti-CD37 modalities.

## 5. Perspectives

CD37 constitutes a promising target for monoclonal antibodies as well as antibody-drug conjugates in B and T cell lymphomas. The major achievements of the recent studies include the generation of DuoHexaBody-CD37 (GEN3009) and employment of CD37-directed CAR-T cells. These two treatment modalities seem to hold the greatest potential for employment in the therapy of relapsed and refractory B cell NHL patients. The promising preclinical results with DuoHexaBody-CD37 have demonstrated its improved efficacy in terms of engaging different mechanisms of the immune response against the tumor. Recently, a first-in-human trial of GEN3009 has been registered for administration in patients with relapsed and refractory B cell NHL (NCT04358458). Promising findings of CD37 as a novel therapeutic target for CARs in the treatment of B cell lymphoma have also been provided [75,85]. The results of the two research papers encourage the development of CD37 CAR-Therapy in the clinical setting. Recently, a study on CD37 CAR-T cells application in hematologic malignancies has been registered and is currently recruiting patients (NCT04136275). The engagement of CD37 in signaling pathways controlling B cell survival encourages further research exploiting it’s targeting together with the use of specific small-molecule inhibitors, as illustrated by anti-CD37 plus idelalisib efficacy [50,69].

However, the use of CD37 as a therapeutic target for immunotherapies raises some important questions. As CD37 has been demonstrated as a negative regulator of tumorigenesis in B cell lymphoma [45], it is important to investigate if anti-CD37 immunotherapy does not lead to the selection of CD37-deficient clones with increased proliferation/metastasis potential.

By now, CD37 is the only tetraspanin targeted therapeutically in humans. It is mostly due to the fact that CD37 expression is restricted to B- and T-lymphocytes, with high expression in B- and T cell-derived tumors. The levels of CD37 on normal T cells are significantly lower than on target cells, which results in the lack of cytotoxicity of naratuximab emtansine against effector T and NK cells in whole blood assays [80], unchanged levels of T cells in otlertuzumab-treated patients [89] and no fratricidal killing when using CD37-CAR-T cells [75]. While CD37 is predominantly being examined for dual targeting for B cell malignancies, the expression of CD37 in some cases of CTCL and peripheral T cell lymphoma (PTCL) makes it an attractive target in these malignancies, otherwise difficult to treat with CAR-T cell-based therapies. The versatility of CD37-CARs to treat both B cell and T cell malignancies indicates that CD37 may be an important target for further investigations. Moreover, the lack of CD37 expression in NK cells provides an opportunity to utilize NK cells as effector cells in CAR-based therapeutic approaches. Although CAR-T therapy was connected with death-leading side effects recognized by cytokine release syndrome or neurotoxicity, it is stated that optimization of the CAR structure or combining CAR-T cell therapy with stem cell transplantation can significantly reduce the chance of facing those adverse effects [90]. However, it is important to bear in mind that CD37 is not a universal target for B cell malignancies, and it cannot be targeted in every B cell malignancy, e.g., not in ALL and multiple myelomas, since the antigen is not present at the beginning of differentiation and becomes reduced in plasma cells [8]. When it comes to other members of the tetraspanin family, also TSPAN33 is highly expressed by activated B cells and in several B cell malignancies [91] and has been proposed to serve as a useful therapeutic target in diseases (B cell malignancies and autoimmune diseases) characterized by an activated phenotype of B cells [91,92]. However, since there are conflicting data on their expression on erythropoietic cells (high expression reported by Heikens et al. [93] vs. low reported by Luu et al. [91]), its employment as a therapeutic target requires further research.

All in all, there is a body of evidence suggesting that CD37 holds potential as a therapeutic target in B cell malignancies, and given its essential role in the development of B cells, the development of approaches combining anti-CD37 immunotherapy with novel small-molecule antineoplastic compounds is warranted.

## Figures and Tables

**Figure 1 ijms-21-09531-f001:**
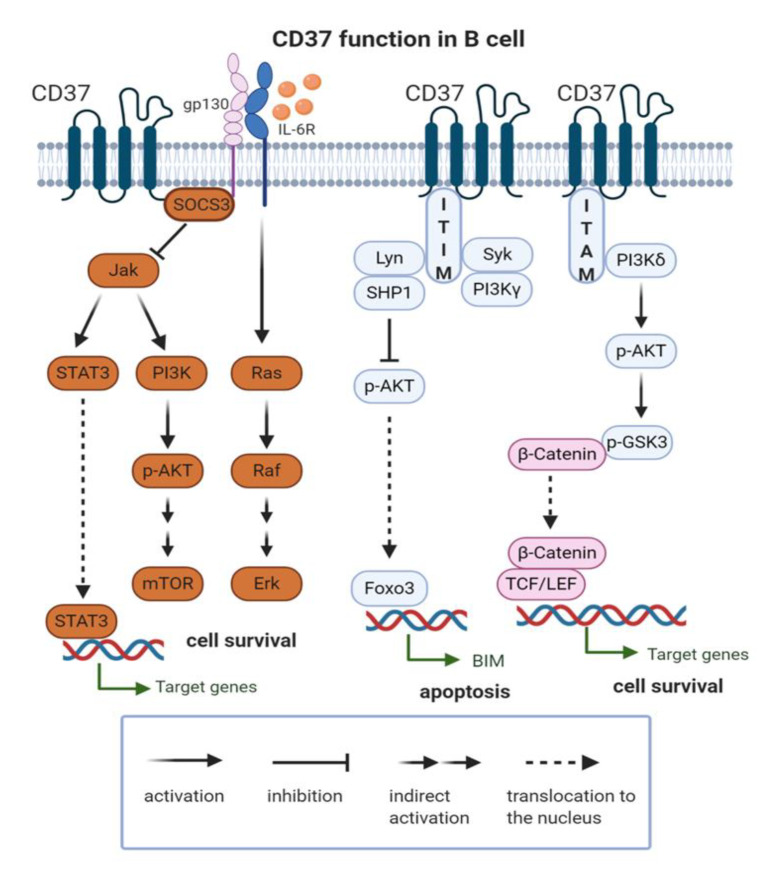
CD37 function in B cell. CD37 molecule is involved in activatory (PI3Kδ/p-Akt/p-GSK3/β-catenin) and inhibitory (SHP1/p-Akt/Foxo3; SOCS3/Jak) signaling pathways. ITAM: immune tyrosine-based activation motif; ITIM: immune tyrosine-based inhibitory motif; IL-6R: Interleukin 6 Receptor; SOCS3: suppressor of cytokine signaling 3; Jak: Janus Kinase; STAT3: Signal Transducer and Activator of Transcription 3; PI3K: phosphatidylinositol 3-kinase; p-Akt: phospho-Akt; mTOR- mammalian target of rapamycin; SHP-1: Src homology region 2 domain-containing phosphatase-1; FOXO3: Forkhead box O3; p-GSK3: phospho-glycogen synthase kinase 3; TCF/LEF: T cell factor/lymphoid enhancer factor.

**Figure 2 ijms-21-09531-f002:**
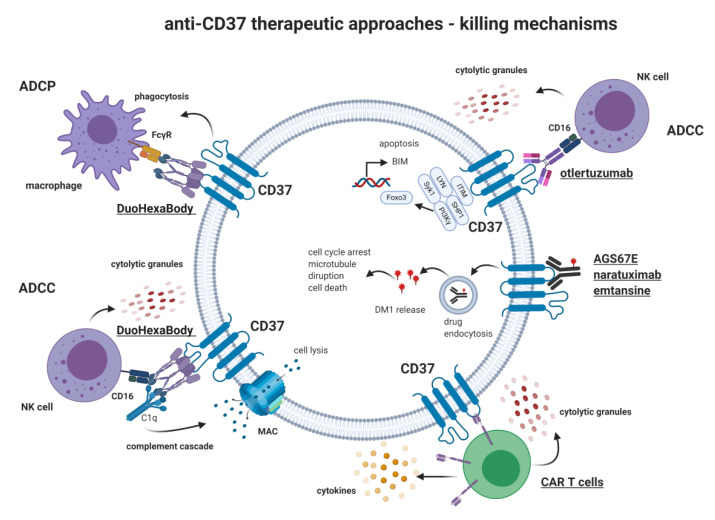
Anti-CD37 therapeutic approaches- killing mechanisms. Several therapeutics have been studied, including monoclonal antibodies (e.g., Otlertuzumab, DuoHexaBody), an antibody-drug conjugate (Naratuximab Emtansine), and chimeric antigen receptor T cells (CAR-T). Different tumor cell killing mechanisms were exploited, e.g., complement-dependent cellular cytotoxicity (CDCC), antibody-dependent cellular phagocytosis (ADCP), antibody-dependent cellular cytotoxicity (ADCC), and direct induction of apoptosis. ITIM: immune tyrosine-based inhibitory motif; MAC: membrane attacking complex; C1q: complement component 1q; FcγR: Fc-gamma receptor; FOXO3: Forkhead box O3.

**Table 1 ijms-21-09531-t001:** Main characteristics of anti-CD37 directed mAbs.

Agent Tested	Basic Characteristics	Mechanism of Action	Clinical Significance
**Otlertuzumab** (formerly known as TRU-016)	Humanized anti-CD37 homodimeric monoclonal antibody engineered with the aid of ADAPTIR platform, based on SMIP-016	Triggers apoptosis dependent upon BIM upregulation in malignant B cells and induces ADCC	Combined administration of otlertuzumab and bendamustine in relapsed CLL patients resulted in 69% overall response. (NCT01188681) [81]
**Naratuximab emtansine** (formerly known as IMGN529)	Humanized anti-CD37 monoclonal antibody conjugated to maytansine-derived microtubule disruptor known as DM1	Induces DM1 internalization and processes it, leading to DM1 intracellular release, disruption of microtubule assembly, and consequently cell cycle arrest and apoptosis; significant ADCC activity observed in preclinical studies	Under investigation in a clinical trial in relapsed or refractory NHL and CLL (NCT01534715)Orphan Drug status (designation for EU–May 2015, for US–September 2016), phase IIb study in relapsed/refractory DLBCL (NCT02564744)
**AGS67E**	Fully human monoclonal anti-CD37 antibody conjugated to the microtubule-disrupting agent MMAE via a protease-cleavable linker	Induces efficient cytotoxicity, apoptosis, and cell cycle alterations following the release of MMAE in numerous NHL and CLL cell lines and patient-derived samples in vitro [70]	AGS67E is characterized by a favorable safety profile, the results of a clinical trial in patients with r/r lymphoid malignancies are awaited (NCT02175433)AGS67E revealed potent activity in B/T cell malignancies and AML in vitro [74]
**BI 836826**	Chimeric Fc-engineered anti-CD37 monoclonal antibody with improved ADCC	Induces apoptosis and demonstrates an improved affinity for the Fc-gamma-RIIIa receptor present on natural killer (NK) cells, leading to effective ADCC	BI836826 demonstrated notable efficacy and acceptable tolerability in phase I clinical trial in patients with CLL (NCT01296932) [5]
**Betalutin** (177) Lu-tetulomab	Murine anti-CD37 monoclonal antibody lilotomab (formerly referred to as HH1) conjugated to the beta-emitting isotope lutetium-177 (Lu-177) via the chemical linker DOTA	Efficiently inhibits cell growth with β-radiation that is emitted from the radionuclide	Under investigation in a clinical trial in relapsed or refractory NHL (LYMRIT-37-05, NCT02658968)

ADCC: antibody-dependent cellular cytoxicity; NHL: non-Hodgkin lymphoma; DLBCL: diffuse large B cell lymphoma; AML: acute myeloid leukemia; MMAE: microtubule-disrupting agent monomethyl auristatin E.

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
