# Peer review of "CD37 in B Cell Derived Tumors—More than Just a Docking Point for Monoclonal Antibodies"

_ijms, 2020, doi:10.3390/ijms21249531_

Round 1
Reviewer 1 Report
The manuscript by Bobrowicz et al provides an interesting overview of the drugs targeting the CD37. The review is globally well structured and touches important points. However some observations representing the critical point of view of the authors as well as some more details are needed to be accepted and understandable and of interest for a broader audience.
Major points
- While the second part of the review (paragraphs 4, 5 and 6) requires only minor revisions, the first part (abstract and paragraphs 1, 2 and 3), in my opinion needs to be restructured.
- I suggest beginning the introduction by presenting the CD37. This could be done by moving the lines from 32 to 38 at the beginning of the introduction. In my opinion the major problem of the introduction is that the authors wanted to make something general (as it should be) but in fact it is not. It provides incomplete information giving a lot of things as assumed (for example the relationship between rituximab and CD37 is not clear at this point as well as which is the competition between CD37 and CD20). Also the sentence “it has been suggested that targeting CD37 with mAbs may be useful for patients resistant or refractory” it is too soon for it in the introduction; at this point it seems just like an assumption as there are no basis to understand this sentence.
I suggest to partially rewrite the introduction being more general and leaving the details for the following paragraphs. As I said good way to start could be to simply present the CD37 by displacing the lines from 32 to 38 at the beginning. And then in the introduction only notions necessary to understand the following paragraphs should be given.
- In my opinion the paragraph 3 is too long, even if it provides important information. To make the review more fluid and understandable I suggest that paragraphs 2 and 3 could be reorganized in one paragraph and rewritten (thus shortening what is currently written) providing a unique presentation of the CD37. And the Figure 1 should be cited in this paragraph.
- I suggest also to add a brief discussion about the “battle with CD20”. Why it has been lost, why CD20 was preferred. If no data are available authors can add their critical point of view. This would provide a better understanding of what is stated afterwards: that also CD37 is a good target.
- If applicable, before moving to paragraph 4 I would add another paragraph describing the relationship of CD37 and rituximab as well as the description of other drugs (if any) whose action passes through CD37. This could be also an occasion of providing a short state of the art of the existing immunotherapies for tumor of B cell origin.
Minor points:
- Not all the drugs listed in table 1 present a reference (e.g. Naratuximab and betalutin)
- In the conclusion I suggest to stress more/summarize why CD37 is a good candidate and which improvement the drugs targeting CD37 could bring to the existing therapies.
Author Response
Reviewer 1
The manuscript by Bobrowicz et al provides an interesting overview of the drugs targeting the CD37. The review is globally well structured and touches important points. However some observations representing the critical point of view of the authors as well as some more details are needed to be accepted and understandable and of interest for a broader audience.
Major points
- While the second part of the review (paragraphs 4, 5 and 6) requires only minor revisions, the first part (abstract and paragraphs 1, 2 and 3), in my opinion needs to be restructured. I suggest beginning the introduction by presenting the CD37. This could be done by moving the lines from 32 to 38 at the beginning of the introduction. In my opinion the major problem of the introduction is that the authors wanted to make something general (as it should be) but in fact it is not. It provides incomplete information giving a lot of things as assumed (for example the relationship between rituximab and CD37 is not clear at this point as well as which is the competition between CD37 and CD20). I suggest to partially rewrite the introduction being more general and leaving the details for the following paragraphs. As I said good way to start could be to simply present the CD37 by displacing the lines from 32 to 38 at the beginning. And then in the introduction only notions necessary to understand the following paragraphs should be given.
Response: We agree with the reviewer and we have accordingly reorganized the introduction part by moving some parts from paragraph 2 and 3 to paragraph 1.
- Also the sentence “it has been suggested that targeting CD37 with mAbs may be useful for patients resistant or refractory” it is too soon for it in the introduction; at this point it seems just like an assumption as there are no basis to understand this sentence.
Response: We decided to keep in the introduction the information about CD37 as a target for immunotherapy with the following sentence: “It seems that CD37 constitutes more than a docking point for monoclonal antibodies and its targeting with immunotherapy may provide additional benefit for refractory patients with B-cell malignancies.”
- In my opinion the paragraph 3 is too long, even if it provides important information. To make the review more fluid and understandable I suggest that paragraphs 2 and 3 could be reorganized in one paragraph and rewritten (thus shortening what is currently written) providing a unique presentation of the CD37. And the Figure 1 should be cited in this paragraph.
Response: We agree with the reviewer and we have shortened paragraph 3 and combined both paragraphs (2 and 3) into one.
- I suggest also to add a brief discussion about the “battle with CD20”. Why it has been lost, why CD20 was preferred. If no data are available authors can add their critical point of view. This would provide a better understanding of what is stated afterwards: that also CD37 is a good target. Figure 1 has now been cited in paragraph 2.
Response: We agree with the reviewer that this part would be extremely valuable. Unfortunately, we cannot encounter any reliable information and data on why CD20 was preferred over CD37 for a relatively long time. We speculate that one of the reasons could be the expression pattern of CD37 and CD20 in B cells and T cells. As shown on the Figure 1 (see attachement), CD37 can be expressed on the surface of healthy T cells at some steps of their differentiation and maturation. From our unpublished data, we also conclude that CD37 can be upregulated on T cells upon their stimulation. As T cells’ depletion by CD37-targeting drugs could be controversial or even detrimental in patients settings, we speculate this could be one of the reasons. However, the CD37 expression non-restricted to B cells could also present the advantage. CD37 target has already been explored in some patients with cutaneous T cell lymphoma and peripheral T cell lymphoma, where it showed good tolerance and partial responses.
- If applicable, before moving to paragraph 4 I would add another paragraph describing the relationship of CD37 and rituximab as well as the description of other drugs (if any) whose action passes through CD37. This could be also an occasion of providing a short state of the art of the existing immunotherapies for tumor of B cell origin.
Response: To make the review concise and for the sake of clarity, we restrained ourselves from describing the existing immunotherapies for B-cell derived tumors. The drugs targeting CD37 have been described in the last paragraph. Regarding the therapies that are indirectly dependent on CD37 expression, we are aware of the role of CD37 in facilitating rituximab action and these observations were described in the fourth paragraph of the review. We are not aware of any other correlations confirmed in clinical trials. Therefore, to avoid any speculations, we also decided not to add this paragraph.
Minor points:
- Not all the drugs listed in table 1 present a reference (e.g. Naratuximab and betalutin)
Response: It has been now corrected.
- In the conclusion I suggest to stress more/summarize why CD37 is a good candidate and which improvement the drugs targeting CD37 could bring to the existing therapies.
Response: We hope that the proposed changes in the text reply to this suggestion. In the text we aimed at citing not only experimental data but also reviews that stress out the importance of anti-CD37 targeted therapies.

Reviewer 2 Report
This is an interesting piece of work, reviewing the role of CD37, mainly as a therapeutic target of relevance, in B-cell malignancies. The review is comprehensive and this topic has not been critically reviewed in the last years, in which new advances have seen the light. Therefore, it presents a novelty in the field and it will be of interest for researchers and clinicians.
I only have suggestions for minor changes in the organization of the manuscript that I think will help the reader follow the message:
-The abstract starts with a sentence explaining what CD37 is. The Introduction should start with a similar sentence.
-The last three paragraphs of section 2 make an introduction to the structure and function of CD37 and tetraspanins. This should be an independent section, which should go right after the Introduction.
-The section CD37 as a molecular target should go after Tetraspanins as immune regulators and could serve as an introductory section to CD37 in B-cell lymphoma and leukemia.
-In Table 1, it would be helpful if the mAbs were presented in the same order as they are in the text. In addition, for Naratuximab, is not there a reference for its orphan drug status in DLBCL?
-The authors should review the manuscript to edit some sentences, since in some places some words seem to be missing, or changes in the punctuation could help better understand some sentences. For instance, in line 188-189, the sentence “As mentioned before, a large... murine models of.” is incomplete.
Author Response
Reviewer 2
This is an interesting piece of work, reviewing the role of CD37, mainly as a therapeutic target of relevance, in B-cell malignancies. The review is comprehensive and this topic has not been critically reviewed in the last years, in which new advances have seen the light. Therefore, it presents a novelty in the field and it will be of interest for researchers and clinicians.
I only have suggestions for minor changes in the organization of the manuscript that I think will help the reader follow the message:
- The abstract starts with a sentence explaining what CD37 is. The Introduction should start with a similar sentence.
Response: We reorganized the paragraphs and now provided the information about CD37 in the introduction part.
- The last three paragraphs of section 2 make an introduction to the structure and function of CD37 and tetraspanins. This should be an independent section, which should go right after the Introduction.
Response: We have reorganized the manusricpt and now the information about the CD37 structure is provided in the introduction, while the CD37 function is depicted in the second paragraph “Characteristics of tetraspanins as immune regulators”.
- The section CD37 as a molecular target should go after Tetraspanins as immune regulators and could serve as an introductory section to CD37 in B-cell lymphoma and leukemia.
Response: We decided first to introduce the role of CD37 in B-cell lymphoma and leukemia, followed by the paragraph about CD37 as a molecular target for immunotherapy. We found this order easier for introducing in the final section information about the CD37-targeting approaches.
- In Table 1, it would be helpful if the mAbs were presented in the same order as they are in the text.
Response: We have accordingly changed the order of the mAbs presented in the table.
- In addition, for Naratuximab, is not there a reference for its orphan drug status in DLBCL?
Response: The information has now been provided.
- The authors should review the manuscript to edit some sentences, since in some places some words seem to be missing, or changes in the punctuation could help better understand some sentences. For instance, in line 188-189, the sentence “As mentioned before, a large... murine models of.” is incomplete.
Response:These corrections have been made.